# *FOXL2* is a Progesterone Target Gene in the Endometrium of Ruminants

**DOI:** 10.3390/ijms21041478

**Published:** 2020-02-21

**Authors:** Caroline Eozenou, Audrey Lesage-Padilla, Vincent Mauffré, Gareth D. Healey, Sylvaine Camous, Philippe Bolifraud, Corinne Giraud-Delville, Daniel Vaiman, Takashi Shimizu, Akio Miyamoto, Iain Martin Sheldon, Fabienne Constant, Maëlle Pannetier, Olivier Sandra

**Affiliations:** 1Université Paris-Saclay, INRAE, ENVA, UVSQ, BREED, 78350 Jouy-en-Josas, France; audrey.lesage.padilla@gmail.com (A.L.-P.); vincent.mauffre@vet-alfort.fr (V.M.); sylvaine.camous@gmail.com (S.C.); philippe.bolifraud@orange.fr (P.B.); Corinne.Giraud-Delville@inrae.fr (C.G.-D.); fabienne.constant@vet-alfort.fr (F.C.); maelle.pannetier@inrae.fr (M.P.); 2Institut Pasteur, UMR 3738, Biologie du Développement et Cellules Souches, Laboratoire de Génétique du Développement Humain, 25 rue du docteur roux, F75015 Paris, France; 3Swansea University Medical School, Swansea University, Singleton Park, Swansea SA2 8PP, UK; g.d.healey@swansea.ac.uk (G.D.H.); i.m.sheldon@swansea.ac.uk (I.M.S.); 4Institut Cochin, INSERM U1016, UMR 8104 CNRS, Faculté René Descartes, 24 rue du Faubourg St Jacques, 75014 Paris, France; daniel.vaiman@inserm.fr; 5Obihiro University of Agriculture and Veterinary Medicine, Obihiro 080-8555, Japan; shimizut@obihiro.ac.jp (T.S.); akiomiya@obihiro.ac.jp (A.M.)

**Keywords:** FOXL2, endometrium, sheep, cattle, progesterone

## Abstract

Forkhead Box L2 (FOXL2) is a member of the FOXL class of transcription factors, which are essential for ovarian differentiation and function. In the endometrium, FOXL2 is also thought to be important in cattle; however, it is not clear how its expression is regulated. The maternal recognition of pregnancy signal in cattle, interferon-Tau, does not regulate *FOXL2* expression. Therefore, in the present study, we examined whether the ovarian steroid hormones that orchestrate implantation regulate *FOXL2* gene expression in ruminants. In sheep, we confirmed that *FOXL2* mRNA and protein was expressed in the endometrium across the oestrous cycle (day 4 to day 15 post-*oestrus*). Similar to the bovine endometrium, ovine *FOXL2* endometrial expression was low during the luteal phase of the oestrous cycle (4 to 12 days post-*oestrus*) and at implantation (15 days post-*oestrus*) while mRNA and protein expression significantly increased during the luteolytic phase (day 15 post-*oestrus* in cycle). In pregnant ewes, inhibition of progesterone production by trilostane during the day 5 to 16 period prevented the rise in progesterone concentrations and led to a significant increase of *FOXL2* expression in caruncles compared with the control group (1.4-fold, *p* < 0.05). Ovariectomized ewes or cows that were supplemented with exogenous progesterone for 12 days or 6 days, respectively, had lower endometrial *FOXL2* expression compared with control ovariectomized females (sheep, mRNA, 1.8-fold; protein, 2.4-fold; cattle; mRNA, 2.2-fold; *p* < 0.05). Exogenous oestradiol treatments for 12 days in sheep or 2 days in cattle did not affect *FOXL2* endometrial expression compared with control ovariectomized females, except at the protein level in both endometrial areas in the sheep. Moreover, treating bovine endometrial explants with exogenous progesterone for 48h reduced *FOXL2* expression. Using in vitro assays with COS7 cells we also demonstrated that progesterone regulates the *FOXL2* promoter activity through the progesterone receptor. Collectively, our findings imply that endometrial *FOXL2* is, as a direct target of progesterone, involved in early pregnancy and implantation.

## 1. Introduction

In mammals, successful implantation and pregnancy depends on a tightly regulated cross-talk between the ovary, the endometrium and the conceptus (embryonic disk and extra-embryonic tissues), which takes place during the peri-implantation period [1,2,3,4]. In cattle, conceptus elongates during the first three weeks of pregnancy, ending with implantation on day 19 to 20 post-*oestrus* [5,6]. During implantation, environmental factors including infections, stress, nutrition or endocrine disruptors can alter this cross-talk leading to early embryonic death [7,8]. In cattle and more specifically in dairy cows, up to two thirds of pregnancies are lost as a consequence of early embryonic death [9,10]. Deciphering the molecular mechanisms of implantation is essential to understand how implantation supports feto-placental development and affects pregnancy outcome [1,3].

Endometrial receptivity and embryo implantation occur during the luteal phase of the oestrous cycle. The receptive status of the endometrium requires combined actions by the ovarian steroid hormones, oestrogen (E2) and progesterone (P4) [11,12,13,14,15,16,17]. Oestrogen is secreted during 4 to 5 days of the follicular phase in cattle [18,19], which is shorter than the 14-day follicular phase in humans (14 days, [18,20,21,22]). In ruminants, the short-term secretion of E2 is not associated with endometrial growth [18], whereas E2 and P4 regulate the proliferation of endometrial cells during the follicular phase in humans, as a prerequisite for invasive implantation [5,22,23]. Progesterone controls the differentiation and secretions of endometrial glands (histotroph; [15,23]) as well as endometrial angiogenesis [15,24]. These processes are essential for ensuring endometrial receptivity, a prerequisite for the establishment and the maintenance of pregnancy as well as conceptus elongation [25,26,27]. The elongating conceptus produces interferon-tau (IFNT), as the major signal of the maternal recognition of pregnancy in ruminants [28,29,30]. Secretion of IFNT prevents regression of the corpus luteum, leading to continued secretion of P4 [31,32]. Despite the importance of P4, the molecular and cellular mechanisms regulated by this hormone are still the object of intense investigation [33,34]. The biological actions of E2 and P4 are mainly mediated by their nuclear receptors, Estrogen Receptor (ESR1-2, formerly named ERα-β, transcribed from two distinct genes) and Progesterone nuclear Receptor (PGRA-B, two isoforms from the same gene; [15,35]), respectively. Upon binding with E2 or P4, the receptors translocate to the nucleus and modulate expression of E2 or P4 target genes [15,35,36,37,38]. In the last decade, studies have identified direct target genes of ESR and PGR in humans, rodents and cattle [39]. These genes are involved in the repression of the E2 signalling pathway (*NR2F2/COUP-TFII*; [38]), proliferation of endometrial cells (i.e., *IGF1* and *EGR1*; [35,38]), uterine receptivity (*FOXO1*, [40]) and decidualisation (*NR2F2/COUP-TFII*, *WNT4*, *HOXA10*; [38]). This non-exhaustive list of steroids targeting genes in the endometrium includes transcription factors that represent regulation nodes of endometrial receptivity and physiology. The biological actions of E2 and P4 are essential for pregnancy outcome but their orchestration is not yet fully elucidated.

In ruminants, transcriptomics has provided a genome-wide understanding of factors involved in endometrial physiology during the oestrous cycle and implantation [25,41,42,43,44,45,46,47,48,49,50]. At the time of implantation, transcriptional regulation of transcription factor expression was reported for several members of the winged-helix/forkhead domain (FOX) transcription factors family in the bovine endometrium [44], including Forkhead Box L2 (*FOXL2*), which appears to be important in the endometrium as well as a key gene involved in ovarian differentiation and maintenance of ovarian function from foetal life to adulthood [51,52,53]. Since our first identification of FOXL2 in the endometrium [52], this transcription factor has been reported to be expressed in endometriotic lesions in humans [54], involved in uterus maturation [55] as well as endometrial cell adhesion with the trophectoderm in mice [56]. However, regulation of *FOXL2* gene expression was IFNT-independent [52] and there was a negative correlation between circulating P4 concentrations and *FOXL2* gene expression in the bovine endometrium [52]. Therefore, the present study aimed to determine if *FOXL2* is a P4 target gene in the endometrium of ruminants. The relationship between P4 and *FOXL2* expression was explored using multiple approaches combining physiological situations in cattle and sheep, in vivo experimental models, as well as in vitro models including FOXL2 promoter analysis in COS cells.

## 2. Results

### 2.1. FOXL2 is Expressed in the Ovine Endometrium 

In sheep, the oestrous cycle lasts 15 to 17 days, and is associated with an increase in P4 secretion (day 5 to 6), which reaches a plateau before decreasing (day 14 to 15) in parallel with corpus luteum regression [25]. 

In sheep, *FOXL2* mRNA and protein were expressed in the caruncular (CAR) and intercaruncular (ICAR) endometrium, and the expression was regulated during the oestrous cycle and implantation (Figure 1). In Western blot analysis, FOXL2 protein was detectable at 50 kDa, as reported previously in cattle [52]. Expression profiles of *FOXL2* were similar for mRNA and protein (Figure 1A,B). Endometrial *FOXL2* expression was significantly higher in CAR areas than in ICAR areas (Figure 1A,B, 3-fold for the mRNA and 5-fold for the protein; *p* < 0.001). During the oestrous cycle, *FOXL2* mRNA and protein expression was higher in early luteal phase (day 4, 1.5-fold) and follicular phase (day 15, 2-fold), compared with the mid-luteal (day 8) and active luteal phase (day 12), as well as at implantation (day 15), especially in CAR areas (*p* < 0.05). 

### 2.2. Endometrial FOXL2 Gene Expression Varies with Blood P4 Concentrations in Sheep

Since endometrial *FOXL2* gene expression decreased when circulating P4 concentrations increased, two ovine experimental models were generated to further analyse the impact of P4 on *FOXL2* endometrial expression. 

Two groups of pregnant ewes were treated with either DMSO solution (control group) or a solution of trilostane, an inhibitor of 3β-HSD, which catalyses the conversion of pregnenolone into progesterone. Trilostane treatment was applied from day 6 to 16 of pregnancy, preventing the P4 concentration increase during the early luteal phase, with a steady concentration of circulating P4 during early pregnancy (Figure 2A). Compared with the control group, *FOXL2* mRNA expression was higher in the CAR endometrium at 16 days of pregnancy in the trilostane-treated group (Figure 2B; 1.4-fold, *p* < 0.05). At the protein level, a trend toward an increase in FOXL2 expression was observed in the CAR areas of the trilostane-treated ewes (Figure 2C, *p* = 0.14).

In order to mimic the ovarian oestrous cycle, ovariectomized ewes were supplemented for 12 days with control solution (control group) or ovarian steroid solutions (P4, E2 or a combination of both) [57]. A group of cyclic ewes at 12 days post-*oestrus* was included in the experimental design in order to determine the effect of ovariectomy on endometrial gene expression. *FOXL2* mRNA and protein were detected in every experimental condition, and both displayed a significant higher expression in the CAR areas compared with the ICAR areas of the endometrium (Figure 3; mRNA and protein, average of 2-fold for mRNA and 3-fold for protein, *p* < 0.05). A significant increase in *FOXL2* transcript expression was observed in the endometrium of the ovariectomized ewes (OVX) compared with the cyclic group (2-fold, *p* < 0.05). Compared with the OVX group, oestradiol (E2) supplementation did not significantly alter *FOXL2* mRNA expression (Figure 3A), whereas FOXL2 protein levels were increased in the ICAR area (Figure 3B; 2.5-fold, *p* < 0.05). Progesterone (P4) supplementation significantly reduced the expression of *FOXL2* mRNA (CAR and ICAR area; 2-fold, *p* < 0.05) and protein (CAR only; 2.5-fold, *p* < 0.05) compared with the control group. In the OVX group supplemented with P4 and E2, *FOXL2* mRNA expression did not significantly differ from *FOXL2* mRNA expression in the OVX, (E2), (P4) and cyclic groups. 

### 2.3. P4 Supplementation Reduces Endometrial FOXL2 Gene Expression in Cattle

Ovariectomized cows were supplemented with E2, P4 or both steroids as described previously [58]. Similar to the ovine OVX model, a significant reduction in endometrial *FOXL2* mRNA expression was detected in (P4)- and (P4 + E2)-supplemented cows, compared with the OVX group (Figure 4A, 2.03-fold and 2-fold respectively, *p* < 0.05). Supplementation with E2 had no significant effect on endometrial *FOXL2* mRNA expression compared with OVX cows. 

Using bovine endometrial explants treated with E2 or P4 for 48h, a significant decrease in *FOXL2* mRNA expression was observed in the P4 treatment group, compared with control or E2 treatment (Figure 4B upper panel, 2.7-fold and 2.3-fold respectively, *p* < 0.05). The expression of *PGR* was also reduced, compared with control or E2 treatment (Figure 4B lower panel, control versus P4 supplementation, 2-fold; P4 versus E2 supplementation, 3.5-fold, *p* < 0.05).

### 2.4. P4 Directly Regulates the Activity of FOXL2 Promoter through Its Nuclear Receptor

Using transiently transfected COS-7 cells, *FOXL2* promoter activity was increased by P4 treatment when PGR-A, PGR-B or a combination of both PGR were overexpressed. (Figure 5, *p* < 0.01; *p* < 0,001). Overexpression of either PGR in the absence of P4 did not significantly modify the *FOXL2* promoter activity.

## 3. Discussion

Since the *FOXL2* gene was cloned in 2001, it has been recognized as a key factor in ovarian differentiation and maintenance of ovarian function from foetal life to adulthood in vertebrates [51,53,59,60,61,62]. However, *FOXL2* is also expressed in the bovine endometrium and its expression was regulated throughout the oestrous cycle and early pregnancy [52]. Endometrial expression of *FOXL2* has been confirmed in several mammals, including mice [55,56], humans [54] and camels [63]. Surprisingly, IFNT, the signal for maternal recognition of pregnancy in ruminants, was not involved in *FOXL2* gene regulation [52]. Therefore, we explored whether ovarian steroids might regulate *FOXL2* expression. In the present study, we demonstrated that P4 regulates *FOXL2* expression in the endometrium of ruminants and stimulates *FOXL2* promoter activity through PGR nuclear receptors. 

In mammals, P4 is essential for initiating and maintaining pregnancy [64]. The present study showed that experimental manipulation of circulating P4 concentrations were associated with variations in endometrial *FOXL2* gene expression in vivo. Ovariectomized ewes or cows supplemented with exogenous P4 had reduced endometrial *FOXL2* expression. This situation was recapitulated in vitro when endometrial explants were incubated with P4. Conversely, *FOXL2* gene expression was up-regulated when the circulating P4 concentrations were low. P4 primarily acts through the nuclear progesterone receptor (PGR), which is involved in uterine receptivity and embryo implantation [15,65,66]. The biological actions of PGR result from regulation of target genes expression when PGR binds to canonical or non-canonical Progesterone Response Element (PRE) motifs present in proximal or distal regions of promoter sequences, modulated by the interactions with co-factors, such as other transcriptional regulators (e.g., ESR1, FOXA1, NCOA3, NCOA1, SRC—String analysis, [67]). In silico analysis of caprine, ovine and bovine *FOXL2* promoter sequences confirmed the presence of a PRE motif that suggests regulation of *FOXL2* gene expression by P4 [68,69]. The association of P4 with PGR affects the expression of numerous transcription factors, including FOXO1, HOXA10 or HAND2 [38], which are involved in cell differentiation and secretory protein production in the uterus [70]. The interaction between P4 and *FOXL2* expression was further evidenced using a COS7 cell line overexpressing PGR protein where P4 supplementation activated the *FOXL2* promoter. Therefore, based on the literature and our data, we hypothesize that *FOXL2* expression is regulated by ovarian steroids E2 and P4 in female reproductive tissues. Further investigations will be necessary to detail mechanisms that drive steroid regulation of *FOXL2* gene expression.

Our initial report [52] and present results demonstrate an inverse correlation between blood P4 concentrations and *FOXL2* gene expression in the endometrium of ruminants. In bovine endometrium, nuclear expression of PGR has been detected in stromal, luminal and glandular epithelia, and the sub-cellular localization of PGR has been shown to vary throughout the oestrous cycle [37]. The nuclear staining was detected during the follicular or early luteal phase when the circulating P4 concentrations were still low (5 days post-*oestrus*). The nuclear signal reduced with the progression of the luteal phase, and became undetectable in luminal and glandular epithelia during the active luteal phase when the P4 concentration was the highest (16 days post-*oestrus*; [37]). Similarly, bovine FOXL2 protein was detectable in the nuclei of endometrial stroma and glandular epithelium during the follicular phase (20 days post-*oestrus*), with the signal intensity decreased in glandular epithelium during the luteal phase (14 days post-*oestrus*; [52]). Our results are consistent with data that reported *FOXL2* as the top gene in a list of 28 transcription factors identified as direct P4-target genes in human umbilical vein endothelial cells (HUVECs) infected with a PGR coding lentivirus [71]. In addition, expression patterns of *PGR* and *FOXL2* transcripts were similar in our model of ovariectomized cows, as well as during oestrous cycle and early pregnancy (published for *FOXL2* [52] and Appendix A for *PGR*). During the oestrous cycle, it is well established that expression of nuclear PGR is downregulated in stromal and glandular cells of bovine endometrium when P4 rises [36,37]. The combination of the present study with former reports in humans [72,73] and cattle [52] indicates that the inverse correlation between P4 plasma levels and endometrial *FOXL2* gene expression does not reflect a negative action of P4 on *FOXL2* gene expression but results from the reduction of PGR expression in endometrial cells. 

Steroid hormones act as critical trophic factors for normal development of numerous biological systems [74]. Oestradiol plays essential roles in female sex determination in non-mammalian vertebrates regardless of the sex determining mechanism [75] and in maintenance of ovarian function in mammalian species [76]. In mice, the combined effect of FOXL2 and ESR2 on granulosa cell identity was demonstrated by genome wide studies that showed common targets shared by FOXL2 and ESR2 [77]. On the other hand, FOXL2 is also known to activate ESR2 expression [77]. Furthermore, FOXL2 stimulates ovarian *CYP19* gene expression, leading to increased E2 production in goats [78], chicken [79], in rainbow trout [80] and medaka fish [81]. These data suggest the existence of a coherent feed-forward loop in which FOXL2 stimulates both oestradiol production and receptivity (i.e., ESR2 expression). In our study, while *FOXL2* transcriptional levels were normal, FOXL2 protein levels were increased by E2 treatment. This finding suggests a stabilization of the protein by post-translational modifications, such as SUMOylation [82] and/or direct interaction with ESR [61]. To our knowledge, this is the first report suggesting a role of oestrogens on FOXL2 protein stabilization, through a feed-back loop. Further analysis will be necessary to elucidate the complex relationship between E2 and FOXL2 protein expression in the endometrium. 

Uterine receptivity is characterized by the intensive proliferation of endometrial cells mediated by E2 during the follicular phase and endometrial gland maturation, changes in endometrial genes expression and P4-mediated decidualisation during the luteal phase in rodents and human [12,13,14,15,16,17,83,84]. In ruminants including cattle, the oestrous cycle is characterized by a short follicular phase and a long luteal phase [18] that are associated with increased and decreased *FOXL2* gene expression, respectively [52]. During the human menstrual cycle, *FOXL2* transcript expression was higher during the follicular phase in keeping with the regulation we reported in ruminants [73]. Altogether, our present and past data as well as other reports have shown that the *FOXL2* gene is expressed during the follicular phase of various mammalian species, suggesting that this transcription factor could be a key regulator of the proliferative process required for establishing uterine receptivity. In granulosa cells, *FOXL2* is involved in the pro-apoptotic process regulating the expression of *BCL2A1* and *ATF3* genes but also the anti-apoptotic process regulating the expression of *TNFAIP3*, *NR5A2* and *FOS* genes [85,86]. Further analyses will be necessary to clarify if *FOXL2* regulates pro- and anti-apoptotic balance as well as cell proliferation in the endometrium during reproductive cycles in mammals.

Since its identification, *FOXL2* gene has been considered as the gatekeeper of ovarian identity due to its highly conserved protein sequence in non-vertebrate and vertebrate species [87]. In mammalian species, new reproductive tissues appeared, including the uterus, placenta and mammary gland. Collectively, our data as well as GEO profiles, NextProt-Beta data and the human protein atlas have documented the expression of *FOXL2* in every female reproductive tissue: the oviduct, uterus (endometrium and myometrium), mammary gland and placenta [52,54,55,56,63,73,88]. Interestingly, whereas *FOXL2* is expressed in each reproductive tissue of mammalian female, its regulation varies with the nature of the reproductive tissue. Our current data have demonstrated that FOXL2 is a progesterone-target gene in the endometrium. In mammalian species, identifying the molecular processes that drive tissue-specific regulation of *FOXL2* gene expression will be mandatory to understand the contribution of this transcription factor to the reproductive process.

## 4. Materials and Methods

### 4.1. Animal Experiments and Cell Cultures

All experimental procedures were completed in accordance with European Community Directive 86/609/EEC and 2010/63/EU and approved by the French Ministry of Agriculture according to French regulations for animal experimentation (authorization number 78-113, approval date 25 September 2006).

Experiment 1. *FOXL2* expression during the oestrous cycle and implantation in the ovine endometrium.

Cyclic and pregnant ewes of the Préalpes-du-Sud breed were synchronized using intravaginal pessaries [89]. Twelve ewes were randomly allocated to three groups (*n* = 4 ewes per group) corresponding to day 4, day 8 and day 12 of the oestrous cycle (representing the early, mid and active luteal phase, respectively). Eight additional ewes were randomly allocated to two groups (*n* = 4 ewes per group) corresponding to day 15 of the oestrous cycle (late luteal phase/follicular phase) and day 15 of pregnancy (implantation). Uteri were collected, flushed, and the stage of pregnancy was confirmed by the presence and morphology of the conceptus in uterine flushings [90]. Endometrial caruncular (CAR) and intercaruncular (ICAR) areas were dissected from the uterine horns ipsilateral to the corpus luteum, as described previously [89]. Endometrial samples were immediately snap frozen in liquid nitrogen, and stored at -80°C prior to analyses. 

Experiment 2. Effect of reduced progesterone concentration on FOXL2 expression in the ovine endometrium.

We investigated the impact of altered circulating P4 concentrations on *FOXL2* expression in ovine endometrium. Pregnant ewes were treated daily for 11 days with trilostane, an inhibitor of 3β-hydroxysteroid-deshydrogenase (3β-HSD) activity, which prevents the conversion of pregnenolone into P4. This ovine experimental model showed that the lower concentration of P4 did not affect conceptus morphology nor pregnancy rates at 16 days post-*oestrus*, but led to changes in endometrial gene expression.

Seventeen pregnant ewes were synchronised, as in Experiment 1. From day 6 to day 16 post-*oestrus*, the pregnant females received either subcutaneous injections of DMSO (*n* = 7) or Trilostane (*n* = 10). Trilostane (15 mg/ewe in 1 mL DMSO) or DMSO was injected every 12 h (08:00 AM and 08:00 PM) into the ewes [91]. Endometrial CAR and ICAR areas were collected and stored as described in Experiment 1.

Experiment 3. Effect of steroid hormone supplementation on *FOXL2* expression in the endometrium. 

### 4.2. In Vivo Supplementation of Steroids in Ewes

Sixteen ewes of the Préalpes-du-Sud breed were ovariectomized (OVX). Then, 42 days after the ovariectomy, they were randomly allocated to five groups (*n* = 4 ewes per group): control ewes (OVX), E2-treated (OVX + E2), P4-treated (OVX + P4) and E2/P4-treated (OVX + E2 + P4) ewes, as described previously [11]. All steroid hormone treatments were administrated in 1 mL of 90% corn oil/10% ethyl alcohol, at intervals of 8 h by intramuscular injection [11]. This steroid hormone administration protocol produces physiological blood concentrations of E2 and P4 [92], corresponding to those during the follicular and luteal phases in cyclic ewes [93]. A control group of 4 cyclic ewes at 12 days of the oestrus cycle was included in the experiment. Blood for monitoring the steroid hormone concentrations and endometrial tissue (CAR and ICAR areas) were collected and treated as described in Experiment 1.

### 4.3. In Vivo Supplementation of Steroids in Cows

Twelve Holstein cows (3 to 7 year of age) were ovariectomized (OVX). Fifty days after the ovariectomy, they were randomly divided into four groups (*n* = 3 cows per group): control cows, OVX (received saline treatment at day 0), OVX + E2 group (received 1 mg of estradiol benzoate (EB) at day 5), OVX + P4 group (the plasma concentration of P4 was increased to the levels of typical mid-luteal using two controlled internal drug release devices (CIDRs) (Pfizer, Tokyo, Japan) inserted into the vagina of animals from day 0 to day 6) and OVX + E2 + P4 group (received two CIDRs at day 0 and 1 mg of EB immediately after removal of the CIDRs at day 6) [58]. Endometrial tissue (ICAR areas) were collected, snap frozen and stored at −80 °C.

### 4.4. Incubation of Bovine Endometrial Explants with Steroids

Endometrial explants were dissected from ICAR areas of the uterus of two cows in the late *oestrus* stage slaughtered at a commercial slaughterhouse, as part of the routine operation of the slaughterhouse [94]. The external surfaces of the uteri were washed in 70% ethanol. The exposed endometrium was washed in Dulbecco’s phosphate-buffered saline solution (D-PBS; Sigma-Aldrich Ltd., Dorset, UK) supplemented with 50 IU/mL penicillin, 50 lg/mL streptomycin (Sigma-Aldrich, Saint-Louis, MO, USA) and 2.5 lg/mL amphotericin B (Sigma-Aldrich). Tissue was collected from the ICAR area of the endometrium using sterile 8-mm-diameter biopsy punches (Stiefel Laboratories Ltd., High Wycome, UK). The explants were cultured ex vivo in 24-well plates (TPP, Trasadingen, Switzerland) containing 2 mL culture medium/well, comprising phenol red-free RPMI 1640 medium (Sigma-Aldrich) containing 10% heat inactivated, double charcoal-stripped, foetal bovine serum (Biosera, East Sussex, UK), as described previously [94,95]. In a former report [94], collection, processing and treatment of explants are described and illustrated with pictures, along with histology using Hematoxylin/Eosin and TUNEL staining to validate the methodology. Within 4 h of slaughter, the explants were treated with control medium, or medium containing E2 (3 pg/mL) or P4 (5 ng/mL) for 48 h. At the end of the culture period, the explants were collected and stored at −80 °C in *Trizol* reagent (Invitrogen, Cergy-Pontoise, France) prior to RNA extraction.

### 4.5. Cell Culture Conditions and Transfections Assays in COS Cells

COS7 cells (Public Health England, Salisbury, UK) were cultured in phenol-red free Dulbecco’s modified Eagle’s media (DMEM)-F12 (Sigma, Saint-Louis, USA) supplemented with 10% (*v*/*v*) heat-inactivated foetal calf serum, 2.52 mM L-glutamine (PAN-Biotech, Aidenbach, Germany), 430-16.4-15.8 nM Insulin-Transferin-Selenium (ITS, Sigma, Saint-Louis, USA), 100 UI/mL- 340 nM Penicillin-Streptomycin (PAN-Biotech, Aidenbach, Germany), 100 UI/mL Nystatin (Sigma, Saint-Louis, USA), 2.7 µM Amphotericin B (PAN-Biotech, Aidenbach, Germany) and 104.7 µM Gentamicyn (Sigma, Saint-Louis, USA). Cells were maintained in 37 °C humidified incubator at 5% CO_2_. 

Cells were plated at a density of 100,000 cells/well in 24-well plates, and were allowed to grow until 90% confluent. After 24 h of steroid deprivation, cells were transfected using 500 ng caprine sequence of the *FOXL2* promoter cloned in pGL3b vector (1 kb proximal promoter, [78,96]), 250 ng pSG5, 250 ng PGRA, 250 ng PGRB or 250 ng PGRA + PGRB (125 ng each), and 10 ng TK-Renillia (internal control for normalisation, Promega Madison, USA) using 1 µL/well XtremeGENE HP DNA Transfection Reagent (Roche Applied Science, Mannhein, Germany). Progesterone Receptor (PGR)-A and -B expressing plasmids, cloned in pSG5 plasmid, were kindly provided by P. Chambon (INSERM, Institut de Chimie Biologique, Faculté de Médicine de Strasbourg, France; [97]). After the 24 h transfection, media were replaced with fresh complete media with 0.03% ethanol (vehicle) or 100 nM progesterone for another 24 h. Luciferase assays were performed with the Promega Dual-Luciferase Reporter Assay System, following the manufacturer’s protocol using the injector Tristar LB941 (Berthold Technologies, Bad Wildbad, Germany), as described previously [78,96]. Each combination of plasmids was tested in duplicate, in three independent experiments. 

### 4.6. Tissue and Cell Collection and RNA Extraction

Total RNA from frozen endometrial tissue and explants was isolated by homogenization using Trizol Reagent (Invitrogen, Cergy-Pontoise, France) as per the manufacturer’s recommendations and purified using Qiagen columns integrating a DNase step (RNeasy mini-kit; Qiagen). 

### 4.7. Real-Time RT-PCR

Total RNA samples were used for real-time RT-qPCR. A total of 1 µg of total RNA were reverse transcribed into cDNA as described previously [52]. Primers (Eurogentec, Lièges, Belgium) were designed (Primer Express Software v2.0, Applied Biosystems) to amplify bovine and ovine *FOXL2* (F: CCGGCATCTACCAGTACATTATAGC; R: GCACTCGTTGAGGCTGAGGT; NCBI sequence reference: NM_001031750.1) and *RPL19* (F: CCCCAATGAGACCAATGAAATC; R: CAGCCCATCTTTGATCAGCTT). Amplified *FOXL2* and *RPL19* PCR fragments were sequenced to assess the amplification of the correct fragment. The expression of *FOXL2* mRNA was calculated and normalized to the reference gene *RPL19* using the relative standard curve method [98].

### 4.8. Western Blot Analysis

Total proteins were extracted from frozen tissue and Western blot immunoassays were processed with 15 µg of total protein extract, as described previously [52], A rabbit anti-FOXL2 purified antibody generated against a peptide corresponding to the C-terminal conserved region of mammalian FOXL2 (WDHDSKTGALHSRLDL, diluted 1:500, CASLO Laboratory, Denmark) was used in PBS-T solution containing 4% non-fat dry milk and then, a goat peroxidase-conjugated anti-rabbit IgG antibody (diluted 1:5000, SantaCruz Biotechnology; Heidelberg, Germany). Actin B protein (ACTB) was assessed as a loading control, using a mouse monoclonal anti-ACTB antibody (diluted 1:2000, A1978—Sigma-Aldrich, Lyon, France) and goat peroxidase-conjugated anti-mouse IgG antibody (SantaCruz Biotechnology; Heidelberg, Germany; diluted 1:5 000). Immunoreaction signals were revealed with Luminata Classico HRP Substrates (Millipore, Guyancourt, France) and analysed using an image analysis system (Advanced Image Data Analyser Software, LAS 1000 camera; Fujifilm, FVST, Courbevoie, France).

### 4.9. Statistical Analyses

Statistical analyses were carried out using GraphPad Prism 6 software (GraphPad Software, USA). Quantitative data were explored using two-way-ANOVA followed by Bonferroni post hoc tests. Data were analysed for effects of day, pregnancy status (cyclic or pregnant), treatments (DMSO and Trilostane; E2, P4 and E2 + P4), endometrial areas (CAR and ICAR) and their interactions (day *versus* status or status *versus* endometrial areas). Quantitative data from the luciferase assays on COS7 cells were explored using one-way ANOVA followed by a nonparametric permutation test, the K-Sample Fisher–Pitman Permutation Test.

## 5. Conclusions

Using various and complementary experimental models carried out with ovine and bovine species as well as in vitro assays; our data have demonstrated FOXL2 as a progesterone-regulated gene in the endometrium of the ruminants. Additional investigation will be necessary to determine if this regulation applies to the endometrium of other mammalian species. Eventually identifying the molecular mechanisms that drive regulation of *FOXL2* gene expression by progesterone in the uterus will provide novel insights about gene regulation by steroids in a tissue-specific manner. 

## Figures and Tables

**Figure 1 ijms-21-01478-f001:**
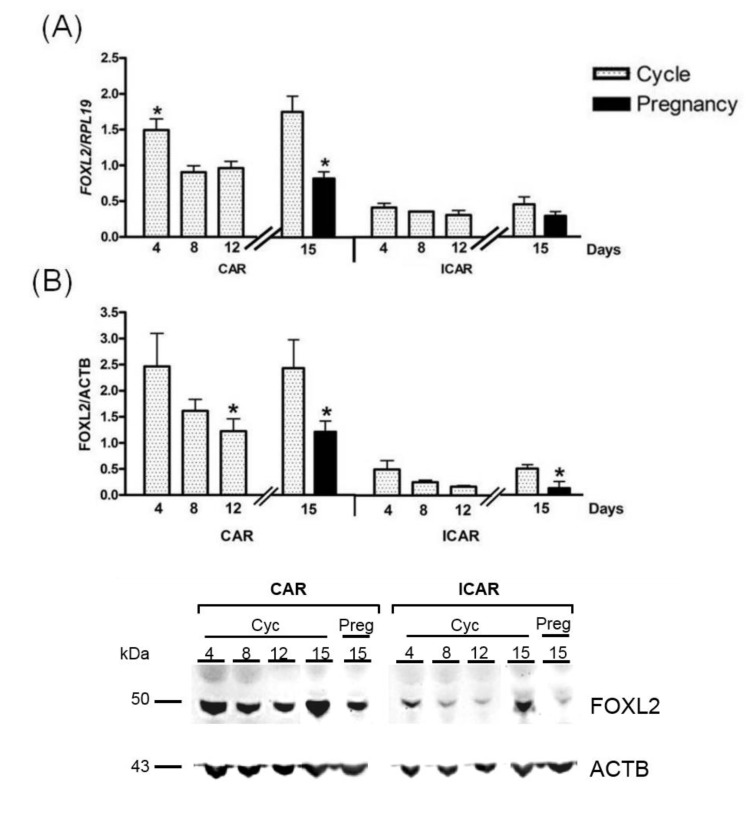
*FOXL2* expression in ovine endometrium. Caruncular (CAR) and intercaruncular (ICAR) endometrial areas were collected from cyclic Pré-alpes du sud ewes at various days post-*oestrus* (day 4, *n* = 4; day 8, *n* = 4; day 12, *n* = 4). Two experimental groups were subsequently added to this study, then CAR and ICAR endometrial areas were collected from cyclic (*n* = 4) and pregnant (*n* = 4) Pré-alpes du sud ewes at 15 days post-*oestrus*. (**A**) Quantification of *FOXL2* mRNA by RT-qPCR. Expression of FOXL2 gene was normalised against that of *RPL19*. (**B**) Quantification of FOXL2 protein by Western blotting. The amount of FOXL2 was normalized to that of ACTB Quantitative data are presented as the mean ± SEM. * *p* < 0.05.

**Figure 2 ijms-21-01478-f002:**
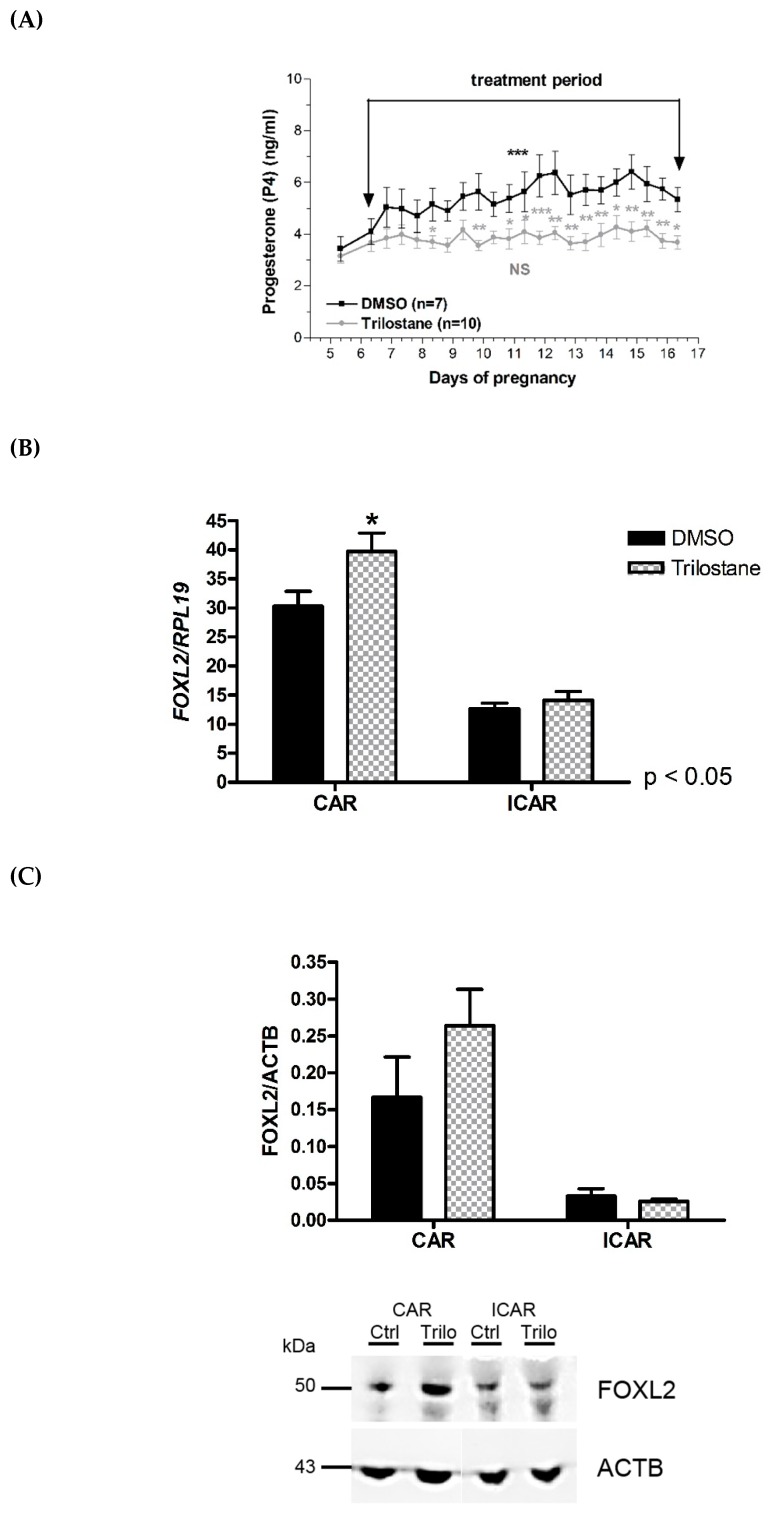
Regulation of *FOXL2* gene expression in ewes with lower circulating P4 concentrations. Caruncular (CAR) and intercaruncular (ICAR) endometrial areas were collected from pregnant Pré-alpes du sud ewes treated with DMSO as a control solution (*n* = 7) or with trilostane (15 mg/ewe in 1 mL DMSO; twice a day, *n* = 10) for 11 days. (**A**) Progesterone dosage throughout the treatment: DMSO (*n* = 7) or Trilostane (*n* = 10) for 11 days. (**B**) Quantification of *FOXL2* mRNA by RT-qPCR. Expression of *FOXL2* was normalized to that of *RPL19.* (**C**) Quantification of FOXL2 protein by Western blotting. The amount of FOXL2 was normalized to that of ACTB. Quantitative data are presented as mean ± SEM. * *p* < 0.05.

**Figure 3 ijms-21-01478-f003:**
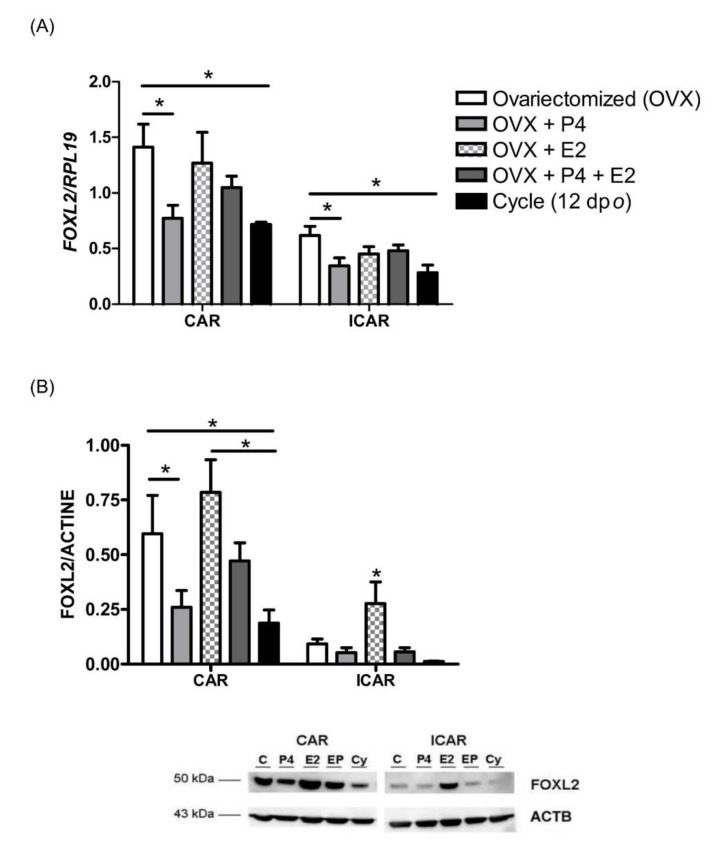
*FOXL2* endometrial expression in ovariectomized ewes supplemented with ovarian steroids. Caruncular (CAR) and intercaruncular (ICAR) endometrial areas were collected from ovariectomized Pré-alpes du sud ewes supplemented with a control solution (OVX, *n* = 4), (P4) solution (OVX + P4, *n* = 4), (E2) solution (OVX + E2, *n* = 4) or (P4 + E2) solution (OVX + E2 + P4, *n* = 4) for 12 days and also from cyclic ewes at 12 days (*n* = 4). (**A**) Quantification of *FOXL2* mRNA by RT-qPCR. Expression of *FOXL2* was normalized to that of *RPL19.* (**B**) Quantification of FOXL2 protein by Western blotting. The amount of FOXL2 was normalized to that of ACTB. Quantitative data are presented as mean ± SEM. * *p* < 0.05.

**Figure 4 ijms-21-01478-f004:**
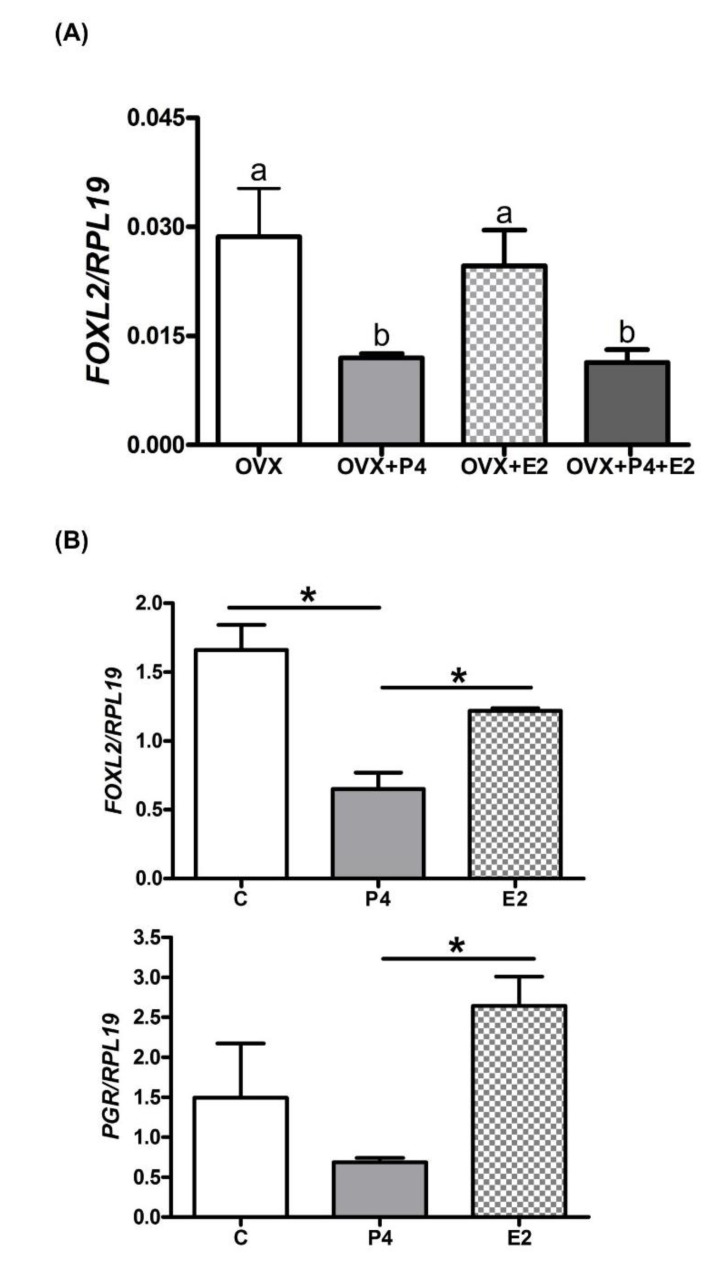
*FOXL2* endometrial expression under the influence of ovarian steroid hormones balance in ovariectomized cows and bovine explants. The expression of *FOXL2* was quantified by RT-qPCR, and normalized to *RPL19.* (**A**) Strips of endometrial tissue were collected from ovariectomized cows supplemented with a control solution (OVX; *n* = 3), progesterone (OVX + P4; *n* = 3), oestradiol (OVX + E2; *n* = 3) or both steroids (OVX + E2 + P4; *n* = 3). Data were analysed by ANOVA and are presented as the mean ± SEM. Bars with different superscripts significantly differ (*p* < 0.05). (**B**) Intercaruncular endometrial explants from two cows were cultured ex vivo for 48 h in control medium (C), or medium containing 5 ng/mL progesterone (P4) or 3 pg/mL oestradiol (E2). The expression of *FOXL2* and *PGR* transcripts was quantified by RT-qPCR and normalized to *RPL19* gene expression. Data were analysed by ANOVA and are presented as the mean ± SEM. * *p* < 0.05.

**Figure 5 ijms-21-01478-f005:**
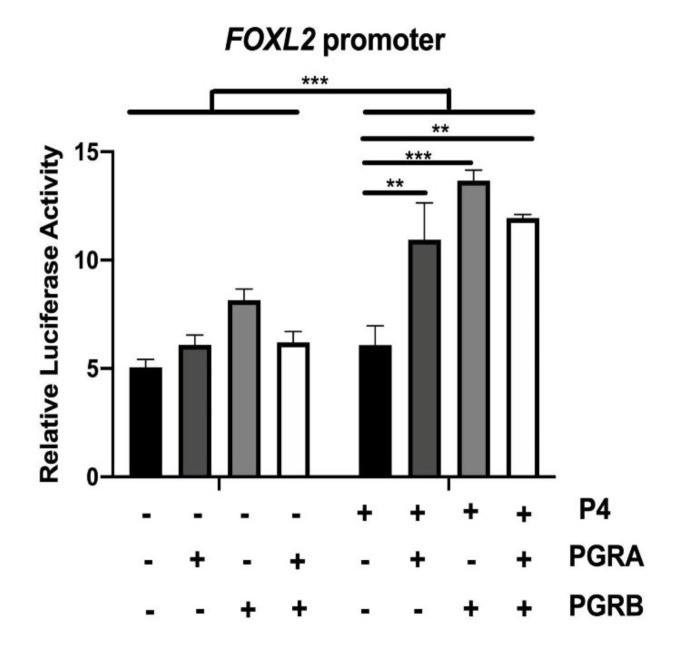
Progesterone regulates *FOXL2* promoter activity in vitro. COS7 cells were cultivated for few passages then transfected using Xtreme gene transfection reagent with progesterone receptor (PGR) A and/or B expressing vectors as well as FOXL2-promoter sequence (1 kb) associated with the luciferase gene for 48 h in the presence or absence of a progesterone (P4) treatment. Activity of the *FOXL2* promoter was normalized to TK–Renilia vector activity. Quantitative data are presented as the mean ± SEM. ** *p* < 0.01; *** *p* < 0.001.

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
