# Peer review of "FOXL2 is a Progesterone Target Gene in the Endometrium of Ruminants"

_ijms, 2020, doi:10.3390/ijms21041478_

Round 1

Reviewer 1 Report

In this manuscript the authors analyze the effect of steroids hormones on the regulation of FOXL2 gene expression in ruminants (cattle and sheep). FOXL2 was expressed in endometrium and was low during luteal phase of estrus and at implantation and it was increased during luteolytic phase. Inhibition of progesterone production by trilostane in pregnant sheep increase FOXL2 in caruncles. Ovariectomized cows or ewes supplemented with exogenous progesterone had inhibition of endometrial FOXL2 expression. Exogenous 17beta-oestradiol in sheep or cattle did not affect FOXL2 endometrial expression and endometrial explants treated with progesterone reduced FOXL2 expression. The authors also demonstrated using in vitro cells assays, that progesterone can regulate FOXL2 promoter activity via its nuclear receptor.   

The work is well done, and indicated that FOXL2 is a direct target of progesterone and could has an important function in endometrial physiology.

Minor points:

Introduction: Indicate the full name for PGR, it is the first time it appears in the text.

-In Fig 1A, no differences were found for FOXL2 expression in pregnancy females, why the authors did not use ACTB for qPCR quantification?

-Fig. 2c should be Fig 2a, because it is the first figure discussed in the text. Again, it is a bit confusing that the RNA results do not match the protein results, if it is a housekeeping problem. Why E2 increase FOXL2 protein levels in the ICAR area? Look like there are some positive effect compared with cycle females.

-Fig 3B, it has been reported that progesterone induces PGR expression in reproductive tract (Steroids 2016, 114:48-58) or it has no effect on PGR mRNA expression (Reproduction 2010, 140: 143-153) , however in the Fig 3b the expression of PGR was reduce when explants were treated with P4.

Author Response

Comments and Suggestions for Authors

In this manuscript the authors analyze the effect of steroids hormones on the regulation of FOXL2 gene expression in ruminants (cattle and sheep). FOXL2 was expressed in endometrium and was low during luteal phase of estrus and at implantation and it was increased during luteolytic phase. Inhibition of progesterone production by trilostane in pregnant sheep increase FOXL2 in caruncles. Ovariectomized cows or ewes supplemented with exogenous progesterone had inhibition of endometrial FOXL2 expression. Exogenous 17beta-oestradiol in sheep or cattle did not affect FOXL2 endometrial expression and endometrial explants treated with progesterone reduced FOXL2 expression. The authors also demonstrated using in vitro cells assays, that progesterone can regulate FOXL2 promoter activity via its nuclear receptor.   

The work is well done, and indicated that FOXL2 is a direct target of progesterone and could has an important function in endometrial physiology.

Minor points:

Introduction: Indicate the full name for PGR, it is the first time it appears in the text.

Response: Progesterone nuclear receptor (PGR) has now been indicated in the introduction (line 78-79).

-In Fig 1A, no differences were found for FOXL2 expression in pregnancy females, why the authors did not use ACTB for qPCR quantification?

Response: We thank the reviewer for raising this specific point. In endometrium, ACTB transcript expression was variable across physiological stages whereas RPL19 was not. This latter was used as the housekeeping gene for transcript expression.

-Fig. 2c should be Fig 2a, because it is the first figure discussed in the text.

Response: We apologize for the confusion. Fig2c is now Fig2a (line 261).

Again, it is a bit confusing that the RNA results do not match the protein results, if it is a housekeeping problem. Why E2 increase FOXL2 protein levels in the ICAR area? Look like there are some positive effects compared with cycle females.

Response: Thank you for your comment. We do no think that the discrepancy stems from normalization of expression. Instead, we have hypothesized E2 could stabilize the FOXL2 protein by post-translational modifications. This is described in discussion of the revised manuscript (line 400-405) as follows: 

“While FOXL2 transcriptional levels were normal, FOXL2 protein levels were increased by E2 treatment, suggesting a stabilization of the protein by post-translational modifications (e.g. SUMOylation [91] and/or direct interaction with ESR [71]. To our knowledge, this is the first report suggesting a role of oestrogens on FOXL2 protein stabilization, through a feedback loop. This aspect deserves investigation in other tissues, in particular in ovarian granulosa cells. Further analysis will be necessary to elucidate the complex relationship between E2 and FOXL2 protein expression in the endometrium mainly through the study of the regulation of ESR.”

-Fig 3B, it has been reported that progesterone induces PGR expression in reproductive tract (Steroids 2016, 114:48-58) or it has no effect on PGR mRNA expression (Reproduction 2010, 140: 143-153), however in the Fig 3b the expression of PGR was reduce when explants were treated with P4.

Response: Thank you for your comment. In the first paper you mentioned (Steroids 2016, 114:48-58), in vitro experiments were carried out with endometrial cancer cells and a P4 agonist. The authors observed an increase of PGR expression. In the second paper (Reproduction 2010, 140: 143-153), high P4 advanced the downregulation of PGR protein from mainly the LE. Low PGR expression in animals with elevated P4 has been observed in sheep (Satterfield et al., 2006). A negative correlation between P4 level and PGR expression was reported at both transcript and protein level. Their data are consistent with our data (see supplementary data 1; PGR expression increases only when P4 level is lowered and vice versa). Overall, long term treatment with P4 is known to downregulated PGR expression in endometrial epithelial cells, as a prerequisite for implantation. In our experiment with explants, reduction of PGR expression was not significant when compared with the control group. The significant effect appears when the P4 –treated group was compared with the E2-treated group treatment.

Reviewer 2 Report

The authors demonstrated the relationship between FOXL2 and endometrium in cattle and sheep. However, the demonstrated data and description is not sufficient as the original article. If authors want to be the article published as an original article, more solid data of cell culture, the morphological image should be represented and the description needs to be more accurate and clear for a better understanding of readership. The abstract the part is too focused on the results, the sentence “Although interferon-Tau does not regulate FOXL2 expression, the impact of ovarian steroids is not clear” is not related to the previous sentence and need to be revised as more specific. Many of the Materials and Methods parts are described as previously published. More detailed descriptions are highly recommended for the in-depth understanding of readership. FOXL2 expression in ovine endometrium Caruncular (CAR) and intercaruncular (ICAR) endometrial areas should be analyzed by immunohistochemistry, thus RT-PCR and western blotting have come from total cells. The specificity should be confirmed. In Fig 1 and 2, the immunoblotting images are not clear. It should be replaced with better magnification and band format like Fig 3. Processing and treatment of explants with ovarian steroid hormones in endometrium explants (Fig.4) should be demonstrated as additional images.

Author Response

The authors demonstrated the relationship between FOXL2 and endometrium in cattle and sheep. However, the demonstrated data and description is not sufficient as the original article. If authors want to be the article published as an original article, more solid data of cell culture, the morphological image should be represented and the description needs to be more accurate and clear for a better understanding of readership. The abstract the part is too focused on the results, the sentence “Although interferon-Tau does not regulate FOXL2 expression, the impact of ovarian steroids is not clear” is not related to the previous sentence and need to be revised as more specific. Many of the Materials and Methods parts are described as previously published. More detailed descriptions are highly recommended for the in-depth understanding of readership. FOXL2 expression in ovine endometrium Caruncular (CAR) and intercaruncular (ICAR) endometrial areas should be analyzed by immunohistochemistry, thus RT-PCR and western blotting have come from total cells. The specificity should be confirmed. In Fig 1 and 2, the immunoblotting images are not clear. It should be replaced with better magnification and band format like Fig 3. Processing and treatment of explants with ovarian steroid hormones in endometrium explants (Fig.4) should be demonstrated as additional images.

Response:

- A native English speaker revised the manuscript.

- As suggested by the reviewer, additional details were added in the “Material and Method” section.

- The objective of our paper is to demonstrate that FOXL2 is a progesterone-regulated gene using experimental models in ruminants. In this context, it was essential to confirm that FOXL2 expression data in sheep were similar with cattle at various stages of estrous cycle and pregnancy. Regarding immunochistochemistry with ovine endometrium, tissue fragments were fixed in order to carry out these complementary analyses. Unfortunately a mistake in paraformaldehyde preparation was made and all our attempts to visualize FOXL2 failed. Despite this technical failure, our data confirm similar profiles of endometrial FOXL2 gene regulation in sheep and cattle.

- As suggested by the reviewer, magnification of immunoblotting images has been improved in Figure 1 (line 238) and Figure 2 (line 266).

- Regarding Fig 4., collection, processing and treatment of explants was illustrated with pictures, along with histology using hematoxylin/eosin and TUNEL staining to validate the methodology. Illustrations can be found in Borges AM, Healey GD, Sheldon IM, 2012 Explants of Intact Endometrium to Model Bovine Innate Immunity and Inflammation Ex Vivo. American Journal of Reproductive Immunology 67: 526–539. A sentence has been added on the “Material and Method” section, line 160-162.

Reviewer 3 Report

I read with great interest the Manuscript entitled “FOXL2 is a progesterone target gene in the endometrium of ruminants” (ijms-702510).  

It is an interesting study aimed to examine whether steroids hormones regulate FOXL2 gene expression in ruminants. The paper is well written, has important clinical message, and should be of great interest to the readers of International Journal of Molecular Sciences. The Manuscript can be further expanded and improved, and reference list can be updated by citing recent studies about the topic.

According to my opinion, only a few small improvements are needed, as suggested below:

I suggest a linguistic revision of the manuscript by a native English speaker. The authors have not adequately highlighted the strengths and limitations of their study. I suggest better specifying these points. I suggest discussing the function of non-coding RNAs in endometrial physiology, analysing their role in endometrial pathologies such as endometrial cancer, endometriosis and chronic endometritis, referring to: PMID: 29663566. Accumulating evidence suggests that estradiol-17β induced endometrial changes may play a pivotal role for the onset of endometrial cancer. I would suggest discussing these elements, referring to: PMID: 30037059; PMID: 21802511.

Author Response

I read with great interest the Manuscript entitled “FOXL2 is a progesterone target gene in the endometrium of ruminants” (ijms-702510).  

It is an interesting study aimed to examine whether steroids hormones regulate FOXL2 gene expression in ruminants. The paper is well written, has important clinical message, and should be of great interest to the readers of International Journal of Molecular Sciences. The Manuscript can be further expanded and improved, and reference list can be updated by citing recent studies about the topic.

According to my opinion, only a few small improvements are needed, as suggested below:

I suggest a linguistic revision of the manuscript by a native English speaker.

Response: We thank the reviewer for his/her suggestion. A native English speaker revised the manuscript.

The authors have not adequately highlighted the strengths and limitations of their study. I suggest better specifying these points. I suggest discussing the function of non-coding RNAs in endometrial physiology, analysing their role in endometrial pathologies such as endometrial cancer, endometriosis and chronic endometritis, referring to: PMID: 29663566. Accumulating evidence suggests that estradiol-17β induced endometrial changes may play a pivotal role for the onset of endometrial cancer. I would suggest discussing these elements, referring to: PMID: 30037059; PMID: 21802511. 

Response: We thank the reviewer for the suggestion and references. Our paper has demonstrated a major regulation of P4 on both FOXL2 transcript and protein expression in ruminant endometrium and a minor effect of E2 on FOXL2 protein expression. We agree that regulation of FOXL2 gene expression by E2 needs further investigation but our data do not allow a discussion about the impact of E2 in endometrial cancer. In addition, although the topic is very attractive, roles of non-coding RNAs in endometrial physiology are beyond the scope of the manuscript. The two final sentences of the discussion have been modified to better recapitulate the major finding of our paper and a future way of investigation. (Line 428-431).

Round 2

Reviewer 2 Report

The authors responded to some of the comments suggested by the reviewer, however, the response and suggested data are still not sufficient for publication.  The visualization of FOXL2 expression data should be demonstrated for the solidity of the manuscript and I don't understand how all the preparation has gone wrong during the procedure. It should be prepared by providing at least one picture together with the demonstration of the procedure recently performed by authors.  Regarding Fig.4, the suggested reference seems as a paper previously published by co-authors, however, the publishing year was 2012. Therefore, the procedure recently performed by authors will solidify the manuscript. Once again, the manuscript still needs the demonstration of data as commented by the reviewer, which will lead the manuscript to fit the criteria for publication. 

Author Response

Reviewer 2, round 2

The authors responded to some of the comments suggested by the reviewer, however, the response and suggested data are still not sufficient for publication. The visualization of FOXL2 expression data should be demonstrated for the solidity of the manuscript and I don't understand how all the preparation has gone wrong during the procedure. It should be prepared by providing at least one picture together with the demonstration of the procedure recently performed by authors. Regarding Fig.4, the suggested reference seems as a paper previously published by co-authors, however, the publishing year was 2012. Therefore, the procedure recently performed by authors will solidify the manuscript. Once again, the manuscript still needs the demonstration of data as commented by the reviewer, which will lead the manuscript to fit the criteria for publication.

Response : The fixation process was used for all the ovine samples simultaneously (experiment 1). Only tissues in experiment 1 were treated for IHC. Unfortunately, a Bouin procedure was used, rather than the regular PFA procedure. We regret that failure in our protocol does not allow the presentation of IHC in the ovine tissues. In the additional PDF file, one of the numerous unsuccessful attempts is presented side by side with a bovine endometrial tissue section that was included during the incubation and washing steps. In sections of bovine PFA-fixed endometrial tissue, FOXL2 cell labeling confirms the data we published in 2012 whereas non-specific FOXL2 cell labeling is seen in ovine Bouin-fixed endometrial tissue.

The objective of this paper was to demonstrate that FOXL2 is a true P4-responsive gene in the endometrium. In order to strengthen our set of data, we had access to ovine experimental models complementary to the ones available in cattle (e.g. experiment 2). Since endometrial FOXL2 gene expression was not available in sheep, we run an extra experiment (named experiment 1). When sheep was compared with cattle, this complementary experiment (exp. 1) demonstrates:

  • Similar profiles of FOXL2 transcript expression across physiological stages of oestrous cycle and implantation
  • Similar regulation of FOXL2 protein expression across oestrous cycle and implantation
  • Similar differential expression of FOXL2 gene between caruncles and intercaruncular areas. FOXL2 gene expression being higher in caruncles

Hopefully this experiment confirms similarities in FOXL2 gene regulation between bovine and ovine species. Then our conclusions in term of P4 regulation could be relevantly extended from cattle to sheep. FOXL2 IHC was carried out with bovine endometrium in our pioneer paper that demonstrated FOXL2 gene regulation in the endometrium of a mammalian species (Eozenou et al, 2012). In the figure presented above, bovine endometrium was used as a control and FOXL2 gene expression was identical to our published paper. Then, even if endometrial FOXL2-expressing cells were not identified in sheep, our results with FOXL2-expressing cells in bovine endometrium and transfection of COS cells demonstrate for the first time that FOXL2 is positively regulated by P4 in the endometrium.
